# Entomo-virological investigation in urban forest fragments and intradomiciles during a dengue outbreak in Salinas, MG, Brazil

**Cirilo Henrique de Oliveira**[1,2,3,/+], **Thaynara de Jesus Teixeira**[1,3], **Rudá Mahayana Cordeiro de Barros**[1,3], **Arlei Bispo de Araújo**[1], **Aline Tátila Ferreira**[1,3], **Danielle Costa Capistrano Chaves**[4], **Fabrício Souza Campos**[5], **Luiz Marcelo Ribeiro Tomé**[6,7], **Natalia Rocha Guimarães**[6,7], **Talita Émile Ribeiro Adelino**[6,7], **Felipe Campos de Melo Iani**[6], **Luiz Carlos Júnior Alcantara**[7], **Walter Santos de Araújo**[2,8], **Filipe Vieira Santos de Abreu**[1,3,9/+]

[1]Instituto Federal do Norte de Minas Gerais, Laboratório de Comportamento de Insetos, Salinas, MG, Brasil
[2]Universidade Estadual de Montes Claros, Programa de Pós-Graduação em Biodiversidade e Uso dos Recursos Naturais, Montes Claros, MG, Brasil
[3]Secretaria Municipal de Saúde de Salinas, Instituto Federal do Norte de Minas Gerais, Centro Colaborador de Entomologia, Laboratório de Comportamento de Insetos, Salinas, MG, Brasil
[4]Secretaria de Saúde do Estado de Minas Gerais, Coordenação Estadual de Vigilância de Arboviroses e Controle Vetorial, Belo Horizonte, MG, Brasil
[5]Universidade Federal do Rio Grande do Sul, Instituto de Ciências Básicas da Saúde, Porto Alegre, RS, Brasil
[6]Fundação Ezequiel Dias, Serviço de Virologia e Riquetsioses, Belo Horizonte, MG, Brasil
[7]Fundação Oswaldo Cruz-Fiocruz, Instituto René Rachou, Belo Horizonte, MG, Brasil
[8]Universidade Estadual de Montes Claros, Centro de Ciências Biológicas e da Saúde, Departamento de Biologia Geral, Laboratório de Interações Ecológicas e Biodiversidade, Montes Claros, MG, Brasil
[9]Fundação Oswaldo Cruz-Fiocruz, Instituto Oswaldo Cruz, Laboratório de Insetos Transmissores de Hematozoários, Rio de Janeiro, RJ, Brasil

**BACKGROUND** Mosquitoes (Diptera: Culicidae) are among the most important disease vectors worldwide. Several species exhibit high levels of anthropophily and are frequently found in human dwellings and forest fragments near urban areas.

**OBJECTIVES** In this integrative study combining mosquito collection, viral detection, and ecological analyses, the assemblage of diurnal mosquitoes was investigated across three distinct environments - intradomiciles, and two distinct urban forest fragments (UFFs) - during a dengue outbreak in the city of Salinas, Minas Gerais, Brazil.

**METHODS** Sampled mosquitoes were tested for the presence of dengue, Zika, and chikungunya viruses through real-time quantitative polymerase chain reaction (RT-qPCR).

**FINDINGS** A total of 722 mosquitoes were collected, representing seven genera and 12 species. The most abundant species were *Culex quinquefasciatus* (270/722, 37.4%), *Aedes aegypti* (205/722, 28.4%), *Ae. albopictus* (112/722, 15.5%), and *Ae. scapularis* (110/722, 15.2%). Five of 81 pools tested positive for dengue virus serotype 1 (DENV-1) RNA, all belonging to *Ae. aegypti* species. Phylogenetic analyses of the nearly complete genome of DENV-1 revealed clustering with strains sampled in 2023 from São Paulo State. Mosquito richness and composition differed between environments (houses and urban forests), whereas abundance was similar across all environments.

**MAIN CONCLUSIONS** Important vector species were detected, including *Ae. aegypti*, *Ae. albopictus*, *Cx. quinquefasciatus*, *Ae. scapularis*, *Sabethes albiprivus*, and *Coquillettidia venezuelensis*, associated with the transmission of dengue, oropouche, mayaro, yellow fever, and Venezuelan equine encephalitis viruses. Entomological and virological investigations in urban and peri-urban environments are crucial, as these areas provide shelter and refuge for anthropophilic and opportunistic mosquito species. Our findings underscore a high potential for mosquito-borne disease spillover in these areas.

Key words: mosquitoes - DENV - forest fragments - spillover - arboviruses - *Aedes* - *Culex*

Financial support: FAPEMIG (grants No. APQ-01403-21 and REDE UAI-ARBO-MG program No. RED-00234-23), National Institutes of Health [grant No. U01 AI151698 as part of the United World Arbovirus Research Network (UWARN)]. This work was partially supported by grants from Secretaria Estadual de Saúde de Minas Gerais and Secretaria Municipal de Saúde de Salinas through the Centro Colaborador de Entomologia program.
CHO is a doctoral fellow supported by the FAPEMIG; LMRT and NRG received scholarships (BDCTI-I) from the FAPEMIG through project RED-00234-23; FSC received a scholarship from the CNPq; TERA received a scholarship from the CNPq under the process number 157696/2025-1.
+ Corresponding authors: cirilohenrique15@gmail.com | ⓘ https://orcid.org/ 0000-0002-3734-8735 / filipe.vieira@ifnmg.edu.br | ⓘ https:// orcid.org/0000-0001-9768-4688

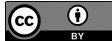

Arboviral diseases are a group of infectious illnesses caused by viruses transmitted by arthropods, primarily mosquitoes (Diptera: Culicidae). Dengue virus (DENV), chikungunya virus (CHIKV), Zika virus (ZIKV), and yellow fever virus (YFV) are notable examples of arboviruses that pose a significant public health challenge worldwide.[1] The clinical manifestations of arbovirus infections range from mild febrile illness to severe neurological, arthritic, and haemorrhagic syndromes.[2]

In recent years, Brazil has experienced multiple outbreaks of arboviral diseases. According to the Ministry of Health,[3] the year 2024 has been marked by a record increase in cases of chikungunya, dengue, and Zika across the country compared to previous years (2021, 2022, and 2023). As of June 2024, the state of Minas Gerais reported 1,637,716 probable cases of DENV (935,899 cases and 733 confirmed deaths), 139,367 probable cases of CHIKV (105,664 cases and 75 confirmed deaths), and 271 probable cases of ZIKV (35 confirmed cases and no deaths).[4]

The species *Aedes aegypti* and *Ae. albopictus* are the main vectors of urban arboviruses such as DENV, ZIKV, and CHIKV.[5] In Brazil, *Ae. aegypti* is a highly anthropophilic species that prefers urban areas, particularly within and around residences. In contrast, *Ae. albopictus* is more commonly associated with rural and peri-urban forested environments.[6,7,8] Urban landscapes are heterogeneous mosaics, interspersing various types of land use and cover, including built areas and highly fragmented vegetation.[9] Forest fragments within the urban environment can provide variable availability of breeding sites, facilitating the maintenance of diverse mosquito species,[10,11] as well as vertebrate hosts for blood-feeding.[12] This dynamic supports the survival and proliferation of anthropophilic and opportunistic/generalist sylvatic mosquitoes. Consequently, this landscape matrix, comprising green areas within or near urban zones, may heighten the risk of spillover of sylvatic viruses (*e.g.*, YFV, oropouche, and West Nile viruses) to humans and spillback of urban viruses such as DENV and CHIKV to wild hosts.[13,14]

Thus, given the need to prioritize more effective strategies for vector monitoring and control, it is essential to understand mosquito species diversity across different habitats and to assess virus infection rates. Accordingly, the objective of this study was to monitor the arboviruses DENV, CHIKV, and ZIKV during a dengue outbreak by analysing the adult mosquito fauna in urban areas and urban forest fragments (UFFs) of Salinas, Minas Gerais. This work represents the first entomo-virological investigation in northern Minas Gerais, that simultaneously integrates molecular arbovirus detection with ecological characterisation of mosquito assemblages in UFFs and intradomiciliary settings during an active dengue outbreak. We also report and phylogenomically analyse the first complete DENV-1 genome from an *Ae. aegypti* population in this region.

## MATERIALS AND METHODS

*Study area* - This study was conducted in the city of Salinas (16º09'45.8″ S; 042º17'54.2″ W), located in northern Minas Gerais (Fig. 1). Salinas has a population of 40,178 inhabitants and a low municipal human development index (MHDI = 0.679).[15] The study area lies in an ecotone between the Cerrado and Atlantic Forest biomes,[16] with the Seasonal Deciduous Forest (commonly referred to as Dry Forest) as its main phytophysiognomy.[17] The region has a semi-arid climate (Aw according to Köppen's classification),[18] characterised by two distinct seasons: a prolonged dry season from March to October and a short rainy season from November to February.

*Mosquito collection* - Collections were conducted between February and June 2024, across three distinct environments — intradomiciles, and two distinct UFFs — during a dengue outbreak. Methods and protocols were previously approved by Brazilian Ministry of the Environment (SISBIO nº 75826-4). Urban mosquito captures were scheduled based on residents' availability, and intradomiciliary mosquitoes were collected and processed as previously described.[19] Briefly, household visits were scheduled according to residents' availability and conducted by a municipal vector surveillance agent and an entomologist equipped with battery-powered Nasci aspirators, oral aspirators, and entomological cages. Sampling involved thorough inspection of all rooms, with special attention to hidden niches such as under furniture and behind cabinets. A total of 36 houses were sampled, with an average collection time of 50 minutes per house, totaling 30 h of collection effort. Peri-urban mosquitoes were captured using the protected human attraction method[20] with entomological nets and Castro aspirators conducted by two collectors simultaneously in two UFFs. The first site (UFF 1) is a riparian urban forest of the Ribeirão stream (Fig. 1), located near a newly developed neighbourhood. This site comprises a small forest fragment connected to a larger fragment, and adjacent to pastures (Fig. 1). The second site (UFF 2) is a riparian urban forest along the Salinas River, situated in the city centre and surrounded by an urban matrix (Fig. 1). A total of 20 h of sampling was conducted in each UFF, amounting to 40 h of total sampling effort.

After capture, mosquitoes were sorted by genus and sex and transferred to field cages using oral aspirators. The cages were then sealed, labelled, and transported to the Insect Behaviour Laboratory at the Federal Institute of Northern Minas Gerais. Mosquitoes were kept alive for three days, as previously described.[19] Subsequently, they were euthanised by freezing at -20ºC, transferred to cryovials, and stored in liquid nitrogen (-196ºC) until further processing.

*Taxonomic identification and molecular diagnosis of captured mosquitoes* - Mosquitoes were transferred from liquid nitrogen and subjected to identification and taxonomic confirmation on a cold table (-20ºC) under a stereoscopic microscope, using dichotomous keys.[21,22] Non-engorged mosquito bodies were pooled (up to 10 individuals) by species and sex. Each mosquito's head was carefully separated from its body under a stereomicroscope, using a sterile scalpel blade dedicated to that specimen to prevent cross-contamination. Mosquito processing and RNA extraction were performed as previously described.[19] Briefly, individual heads and pools of up to 10 bodies from the same species and sex were placed in

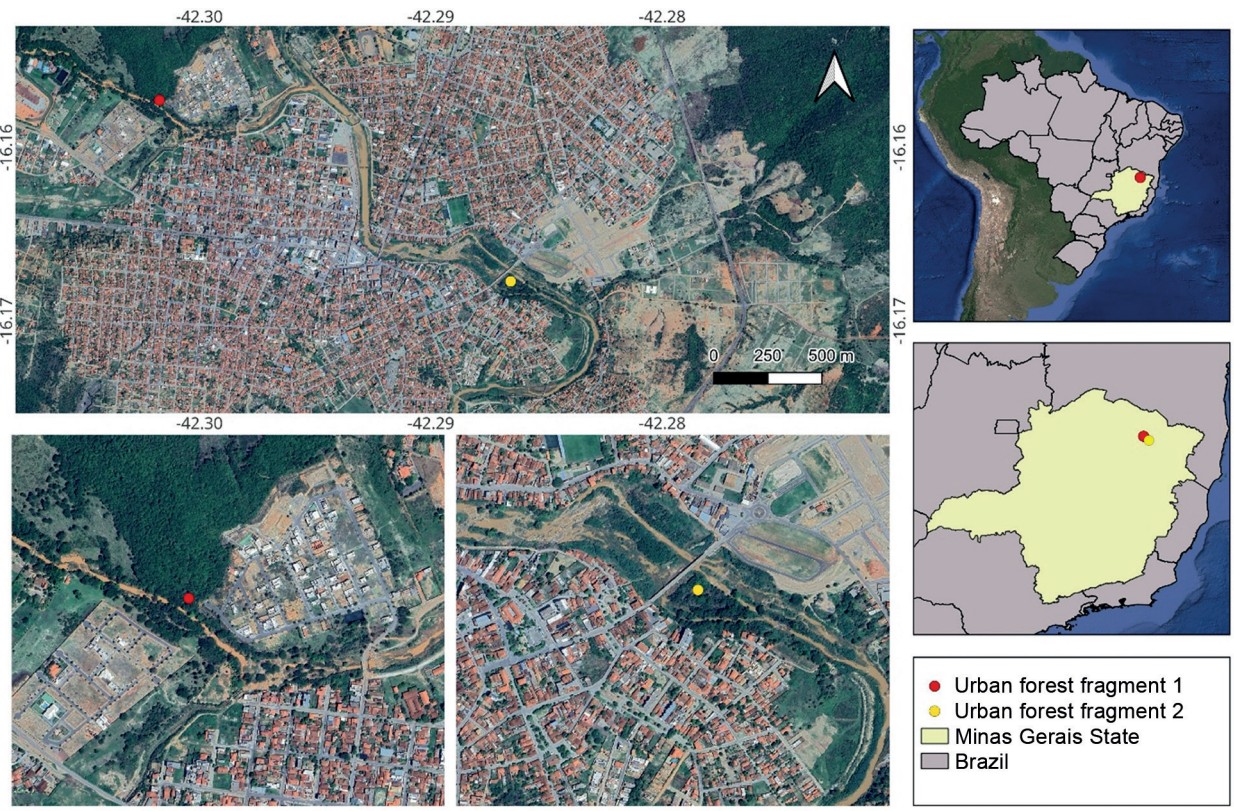

Fig. 1: map showing the city of Salinas, where mosquito captures were conducted. Sampling sites in urban forest fragments (UFFs) are indicated by red and yellow dots. The figure was created using the free software QGIS 3.40.3.

enriched L-15 medium (Thermo Fisher Scientific Cat. No. 11415064; enriched with 20% foetal bovine serum (FBS), 0.5% non-essential amino acids, 1% penicillin, 0.1% gentamicin, and 0.1% fungizone). Tissues were homogenised with beads in a beadbeater for 30 s at 7500 rpm, followed by centrifugation at 12,000 rpm for 8 min at 4ºC. A 140 μL aliquot of the resulting supernatant was used for RNA extraction with the QIAamp Viral RNA Mini Kit (Qiagen, Cat. No. 52906), according to the manufacturer's instructions. RNA from body pools was tested for DENV, ZIKV, and CHIKV by real-time quantitative polymerase chain reaction (RT-qPCR) using the Bioclin ZDC Multiplex One-Step kit (Cat. No. G021-3), with 9 μL of total RNA as input, following the manufacturer's protocol. For DENV-positive pools, RNA from each mosquito head was individually tested to confirm viral dissemination. DENV-positive pools were also subjected to a specific one-step RT-qPCR assay to identify the serotype (DENV-1 to DENV-4) using a previously described protocol, with 5 μL of total RNA as input.[23] All samples were tested in duplicate. Samples were considered positive when the cycle threshold (Ct) was below 40.

*DENV genome sequencing and phylogenomic analyses* - The RNA extracted from the DENV-1 positive pool (X-737) was subjected to cDNA synthesis and PCR amplification following a sequencing protocol based on the multiplex PCR-tiling amplicon approach.[24] The resulting amplicons were then purified using 1× AMPure XP Beads (Beckman Coulter, Brea, CA, USA, Cat. No.

A63880) and quantified with a Qubit 3.0 fluorometer (Thermo Fisher Scientific, Waltham, MA, USA) using the Qubit™ dsDNA HS Assay Kit (Thermo Fisher Scientific, Cat. No. Q32851).

The sequencing library was prepared using the Native Barcoding Kit 96 V14 (SQK-NBD114.96) from Oxford Nanopore. Sequencing was performed on a MinION device (Oxford Nanopore) for 24 h, using the Flow Cell R10.4.1 (FLO-MIN114) and the MinKNOW software (v24.02.16) with basecalling disabled. Basecalling and demultiplexing of the libraries were conducted using Guppy software (v6.5.7).

*Bioinformatics a*nalysis - The genome of *Orthoflavivirus denguei* serotype 1 (DENV-1) was assembled using the reference-based assembly approach, implemented in a custom pipeline. This pipeline integrates the following tools: NanoPlot (v1.43.0, github.com/wdecoster/NanoPlot) for quality assessment, Chopper (v0.8.0) for adapter and low-quality base removal, Minimap2 (v2.28) for alignment, Samtools (v1.20) for alignment file manipulation, Bcftools (v1.20) for variant calling, iVar (v1.4.2) for consensus genome generation, and Pilon (v1.24) for correction and refinement of the final assemblies. The reference sequence was obtained from the NCBI under the accession number NC_001477.1, corresponding to DENV-1. The genome quality assessment and genotyping were performed using Nextclade (v3.10.0). The newly generated DENV-1 genome sequence has been deposited in GISAID under accession number EPI_ISL_19725409.

For the phylogenetic analysis, the assembled genome of sample X-737 was initially aligned with 2,527 DENV-1 genomes obtained from the GISAID database, with collection dates ranging from 2022-11-02 to 2024-12-17 [Supplementary data (Table)]. The alignment was performed using the MAFFT software and was manually inspected in AliView. Subsequently, a preliminary phylogenetic analysis was conducted using FastTree (v2.1.11). Based on the generated phylogenetic tree, the most representative clade containing the X-737 genome was identified. From this clade, a subset of sequences was selected, resulting in a set of 619 DENV-1 genomes, which were aligned with the X-737 genome and inspected as previously described. The final alignment file was used to infer the most suitable evolutionary model and to perform the maximum likelihood (ML) phylogenetic analysis using IQ-TREE2 (v2.3.6). The statistical robustness of the tree topology was assessed using 1000 bootstrap replicates.

*Ecological analysis* - Generalised linear models (GLMs) were constructed to test the effect of the environment — houses, UFF 1, and UFF 2 (explanatory variables) — on mosquito richness and abundance (response variables). To assess mosquito abundance, we employed GLMs with a negative binomial distribution due to overdispersion in the data. For species richness, we adopted GLMs with Poisson distribution. All models were subjected to residual analysis using the testDispersion function from the "DHARMa" package[25] to assess the adequacy of the error distribution.[26] Subsequently, all models were analysed using analysis of variance (ANOVA), with significance assessed using the $\chi^2$ test for the abundance model and the F test for the richness model.[26] To determine whether the areas differed significantly from each other, contrast analysis was also performed. Non-Metric Multidimensional Scaling (nMDS) based on the Bray-Curtis similarity index was used to compare species composition between urban, and two forest fragments environments. A non-parametric permutation procedure (ANOSIM) with 999 permutations, also based on the Bray-Curtis similarity index, was then applied to test the significance of the groups formed in the nMDS. All statistical analyses were performed using R software.[27]

## RESULTS

*Species collected and infection rates* - A total of 722 mosquito specimens were collected, representing 12 taxonomic units, with 10 taxa identified at the species level. Of the 395 mosquitoes collected in the urban environment (intradomiciles), all belonged to the subfamily Culicinae, tribes Aedini, and Culinici, comprising only two genera: *Culex* and *Aedes*. The most abundant species was *Culex quinquefasciatus* (n = 203), followed by *Ae. aegypti* (n = 191) and *Ae. scapularis* (n = 1). In the UFFs (UFF 1 and 2), the 327 specimens were distributed across seven genera and 12 taxa. The most abundant species were: *Ae. albopictus* (n = 112; 34.3%), *Ae. scapularis* (n = 109; 33.3%), and *Cx. quinquefasciatus* (n = 67; 20.5%) [Table I and Supplementary data (Table)]. *Ae. aegypti* and *Wyeomyia* sp. were exclusively detected at UFF1, while *Mansonia humeralis*, *Ae. serratus*, *Coquillettidia venezuelensis*, *Psorophora albipes*, and *Cx. (Melanoconion)* spp. were restricted to UFF2 [Supplementary data (Table)].

The mosquito bodies were grouped into 81 pools. All pools tested negative for CHIKV and ZIKV. Notably, five *Ae. aegypti* pools (comprising 28 mosquitoes) tested positive for DENV-1, as did six individual heads, with CT values ranging from 26.4 to 28.4 (Table II).

TABLE I

Species, number, relative abundance, and infection rate of mosquitoes sampled

| Species | Number of mosquitoes | | | Relative abundance (%) | No. tested - Pools tested (DENV positive pools) | MIR |
| | House (%) | UFF (%) | Total | | | |
|---|---|---|---|---|---|---|
| *Culex quinquefasciatus* Say, 1823 | 203 (51.4) | 67 (20.0) | 270 | 37.4 | 260 - 25 (0) | 0 |
| *Aedes aegypti* (Linnaeus, 1762) | 191 (48.3) | 14 (4.3) | 205 | 28.4 | 179 - 35 (5) | 27.9 |
| *Aedes albopictus* (Skuse, 1894) | 0 | 112 (34.3) | 112 | 15.5 | 64 - 7 (0) | 0 |
| *Aedes scapularis* (Rondani, 1848) | 1 (0.3) | 109 (33.3) | 110 | 15.2 | 104 - 11 (0) | 0 |
| *Psorophora ferox* (Humboldt, 1819) | 0 | 9 (2.8) | 9 | 1.2 | 5 - 2 (0) | 0 |
| *Culex (Mel.)* spp. Theobald, 1903 | 0 | 7 (2.1) | 7 | 1.0 | 0 - 0 (0) | 0 |
| *Mansonia humeralis* Dyar & Knab 1916 | 0 | 3 (0.9) | 2 | 0.4 | 2 - 1 (0) | 0 |
| *Sabethes albiprivus* Theobald, 1901 | 0 | 2 (0.6) | 2 | 0.3 | 2 - 0 (0) | 0 |
| *Aedes serratus* (Theobald, 1901) | 0 | 1 (0.3) | 1 | 0.1 | 0 - 0 (0) | 0 |
| *Coquillettidia venezuelensis* (Thobald, 1912) | 0 | 1 (0.3) | 1 | 0.1 | 0 - 0 (0) | 0 |
| *Psorophora albipes* (Theobald, 1907) | 0 | 1 (0.3) | 1 | 0.1 | 0 - 0 (0) | 0 |
| *Wyeomyia* sp. Theobald, 1907 | 0 | 1 (0.3) | 1 | 0.1 | 0 - 0 (0) | 0 |
| Total | 395 (54.3) | 327 (45.7) | 722 | 100.0 | 614 - 81 (5) | - |

DENV: dengue virus; MIR: minimum infection rate = number of positive pools / number of same species adults analysed × 1000; UFF: urban forest fragment (the sum of UFF 1 and 2).

*DENV genome sequencing and phylogenomic analyses* - Sequencing of sample X-737 generated an *O. denguei* genome with 93.60% coverage and an average depth of 2,150X (26,334 reads mapped to the reference genome). Genome quality assessment, conducted using the online tool Nextclade, confirmed the high quality of the genome and classified it as dengue virus serotype 1 (DENV-1), genotype V. Maximum likelihood (ML) phylogenetic analysis revealed that the newly generated genome clustered with genomes from the Midwest and Southeast regions. Specifically, it was more closely related to two genomes from São Paulo (Southeast region), collected in February and April 2023 (Fig. 2).

*Ecological analyses* - Species richness differed significantly (F = 3.626; P = 0.026) among the treatments (environments), with houses exhibiting lower species richness compared to forest environments (Fig. 3). In contrast, mosquito abundance showed no significant difference between environments (Chisq = 34.759; p = 0.507).

Significant differences in mosquito assemblage composition were observed across the environment types (ANOSIM: R = 0.510; P = 0.001), with the most pronounced difference occurring between houses and UFFs. This is visually evident in the NMDS diagram, which shows a distinct grouping of samples by environment type (Fig. 4).

TABLE II

Description of dengue virus serotype 1 (DENV-1)-positive pools

| Cod. pool | Species | No. of individuals | Sex | Cycle threshold (Ct) | Positive heads |
|---|---|---|---|---|---|
| X723 | *Aedes aegypti* | 7 | F | 27.7 | ITc417; ITc421 |
| X733 | *Ae. aegypti* | 6 | F | 27.6 | ITc467 |
| X737 | *Ae. aegypti* | 5 | F | 26.4 | ITc487 |
| X757 | *Ae. aegypti* | 5 | F | 26.5 | ITc554 |
| X760 | *Ae. aegypti* | 5 | F | 28.4 | ITc567 |

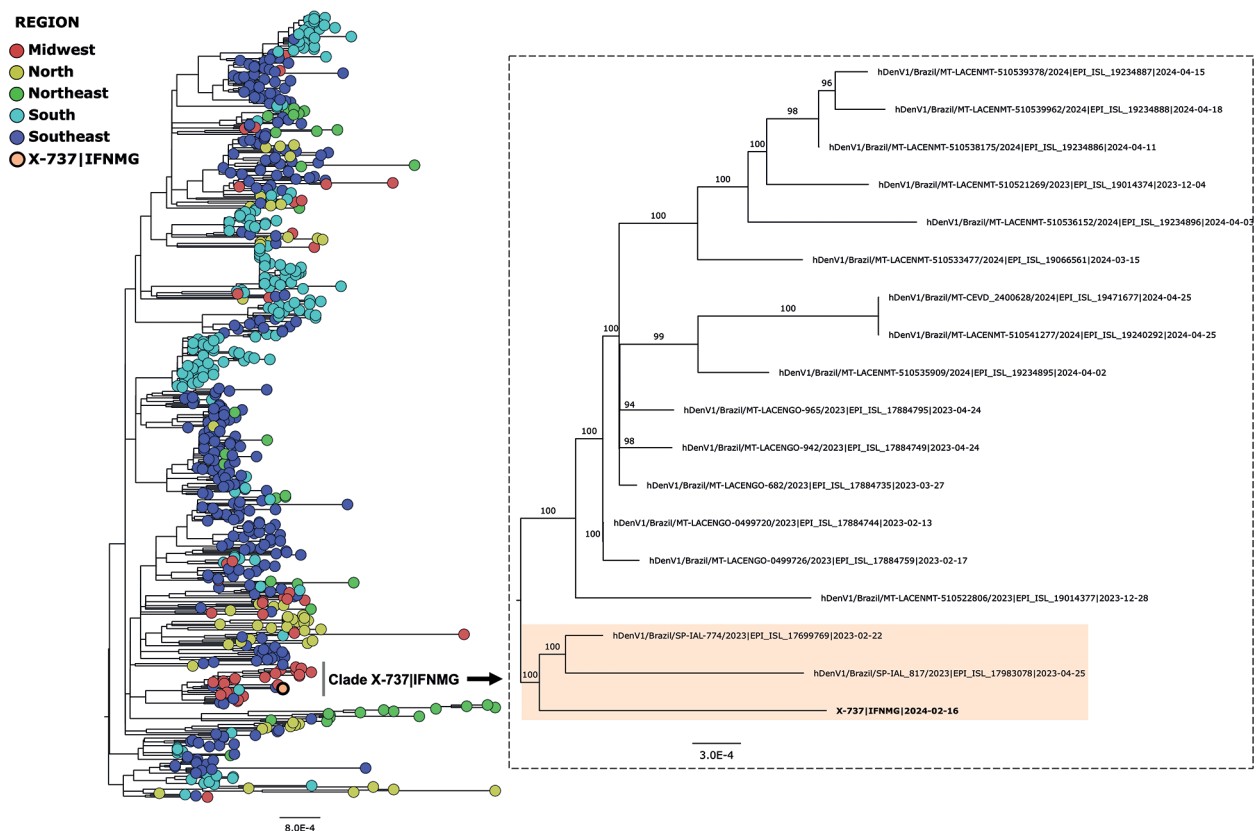

Fig. 2: maximum likelihood (ML) phylogenetic tree generated in IQ-TREE using the TIM2+F+I+R5 evolutionary model with 1000 bootstrap replicates. The tree was constructed using a dataset of 619 dengue virus serotype 1 (DENV-1) genomes from the GISAID database, along with the genome generated in this study from sample X-737. In the phylogenetic tree on the left, the genome generated in this study is highlighted, showing the clade in which it clustered. On the right, an expanded view of the subclade containing the X-737 genome is displayed, with the genomes highlighted in orange.

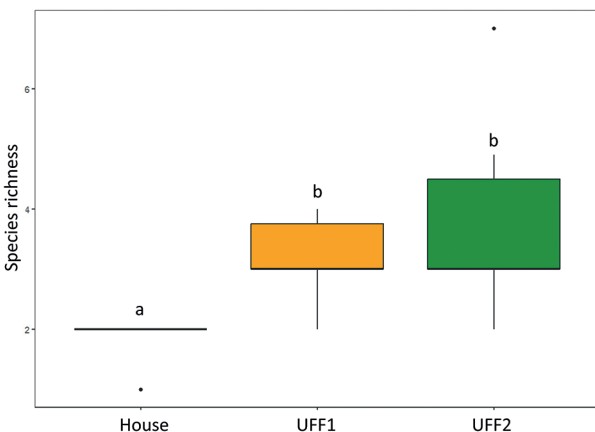

Fig. 3: comparison of mosquito species richness among houses, urban forest fragment 1 (UFF1) and UFF2. Different letters indicate significant differences between the bars.

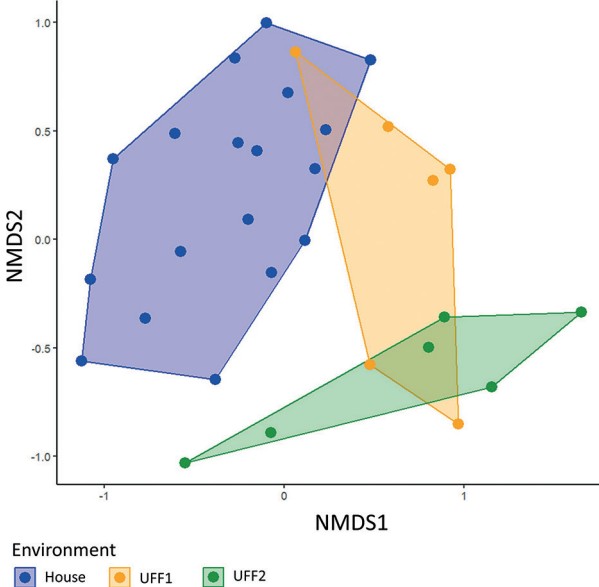

Fig. 4: non-metric multidimensional scaling (nMDS) showing the ordination of mosquito species composition across house, urban forest fragment 1 (UFF1) and UFF2 environments.

## DISCUSSION

The urban and peri-urban mosquito fauna in Salinas, northern Minas Gerais, was investigated during the largest dengue outbreak in recent years. We identify *Ae. aegypti* mosquitoes, as the primary vector of DENV-1, with no evidence of other species implicated in the transmission of this virus. Most entomo-virological surveillance has focused on highly urbanised settings, particularly households, and targeted only the main vectors, *Ae. aegypti* and *Ae. albopictus*.[28,29,30,31] In contrast, studies in UFFs have mainly described mosquito fauna without virological screening[32,33,34] or conducted entomo-virological monitoring during periods without active viral circulation.[35] Our study addresses this gap by jointly assessing mosquito assemblages and arbovirus circula-

tion in both households and UFFs, emphasising their relevance for detecting potential spillover/spillback events and for strengthening early warning systems.

Dengue is currently the most significant mosquito-borne viral disease globally.[36,37] In 2024, between epidemiological weeks 1 and 34, approximately 6,500,000 probable cases of dengue were reported in Brazil, with 5,244 confirmed deaths attributed to the disease.[38] DENV consists of four antigenically distinct serotypes, capable of triggering explosive outbreaks of varying magnitudes.[39,40] Both DENV-1 and DENV-2 were circulating throughout the state of Minas Gerais in 2024, causing human cases.[4,41] Among the infected mosquitoes, we identify only the DENV-1 serotype, which predominated in this outbreak, including in the study region.[4] The clade formed by the DENV-1 genome sequenced from Salinas (MG) and two genomes from São Paulo (SP), collected in February and April 2023, possibly reflects the high rate of human migration from northern Minas Gerais to São Paulo in search of work, as well as the absence of DENV-1 genomes from local human cases, due to limited infrastructure.

The high frequency of *Cx. quinquefasciatus* and *Ae. aegypti* indoors is consistent with the primarily urban and anthropophilic habitat of these vectors.[21] However, despite the higher abundance of *Cx. quinquefasciatus*, only *Ae. aegypti* was found naturally infected with DENV-1, including viral detection in the head, indicating virus dissemination throughout the mosquito's tissues. Notably, this is the first DENV-1 genome sequenced from an *Ae. aegypti* population in Minas Gerais. The minimum infection rate (MIR) for *Ae. aegypti* observed in this study is higher than those previously reported in the literature (MIR = 1.47 for adult *Ae. albopictus* collected in spring 2014 in São Paulo, Brazil, and 3.37 and 16.2 for immature and adult *Ae. aegypti*, respectively, collected in Rio Grande do Norte, Brazil, between 2011 and 2014).[42,43] The high indoor abundance of mosquitoes, combined with the MIR (27.9) observed in *Ae. aegypti*, underscores the role of this species in maintaining and transmitting DENV in Brazilian urban environments.[42]

Although the primary dengue vector in the Americas is the urban mosquito *Ae. aegypti*, *Ae. albopictus* exhibits high vector competence and has been found naturally infected with DENV (including DENV-1) in Brazil.[42,44] Albeit the presence of *Ae. albopictus* in ovitraps in the peridomestic areas of Salinas has been documented,[45] this species was not found inside the sampled domiciles. In Brazil, *Ae. albopictus* typically exploits forest edges in transitional areas (ecotones) between forests and urban landscapes, positioning it as a potential bridge vector for arboviruses between these environments.[46] This behaviour aligns with our findings, as *Ae. albopictus* was the most abundant species in the sampled UFFs. Interestingly, one specimen of *Ae. scapularis* was found inside one sampled domicile. While the presence of *Ae. scapularis* indoors may be incidental, this species has previously been documented indoors during a CHIKV outbreak[47] and has been found naturally infected with YFV.[48,49] Both *Ae. albopictus* and *Ae. scapularis* are potential arbovirus vectors, exhibiting ecological ver-

satility, adaption to anthropogenic environments, and eclectic blood-feeding behaviour.[22,50,51,52,53] Therefore, these species warrant attention, as they can feed on a wide variety of vertebrates found in urban green areas, in addition to humans, potentially acting as bridge vectors for zoonotic transmission.[12,21]

Urban green spaces — while essential for ecosystem services and human well-being — also constitute pivotal ecological niches that influence arbovirus dynamics. In forest edges and UFFs, the co-occurrence of vector species underscores such areas' potential as bridge zones linking sylvatic and peri-urban transmission cycles.[46,54] In the Brazilian Amazon, the detection of ZIKV in *Ae. aegypti* and *Ae. albopictus* collected within UFFs further highlights how such green spaces may facilitate pathogen circulation.[31] Urban greening efforts must be paralleled by vigilant, ecologically informed arbovirus monitoring to effectively mitigate human health risks. Here, the mosquito fauna detected in the small UFFs suggests that these urban green spaces, even when heavily modified by human activity, provide favourable environments for the persistence of medically important mosquito species. Beyond *Ae. scapularis* e *Ae. albopictus*, other potential vectors identified include *Ps. ferox* and *Sabethes albiprivus*, which have been previously found naturally infected with YFV[55,56] and *Cq. venezuelensis*, a vector of oropouche and Venezuelan equine viruses.[57,58] Casual collections included species highly adapted to urban environments, such as *Cx. quinquefasciatus* and *Ae. aegypti*. The persistence and abundance of these vectors in urban green areas underscore their opportunistic behaviour and reflect the ongoing process of anthropisation.

We observe differences in species richness and composition, while species abundance was similar across the three environments (intradomicile, UFF1, and UFF2). Previous studies have also shown that mosquito fauna can vary between indoor and outdoor environments associated with households.[59,60,61,62] Although it is beyond the scope of this study, differences in species richness and composition are likely influenced by the association of mosquitoes with landscape types.[63] Heterogeneous landscapes support greater mosquito diversity compared to simplified landscapes, as they provide a higher availability of breeding sites[11] and a greater abundance of food resources for mosquitoes.[12,64] The houses and UFFs shared three species, among which *Cx. quinquefasciatus* was the most abundant. The high abundance of *Cx. quinquefasciatus* in urban environments is directly related to its ability to breed in polluted water bodies rich in organic matter.[65] Such water collections were observed in the streets as well as in an open sewage channel near the UFF1 sampling site. Finally, the exclusive presence of five sylvatic mosquito species in UFF2 highlights the ability of small forest fragments, even when surrounded by urban landscapes, to sustain diverse vector populations.

The absence of DENV, CHIKV, and ZIKV detection in mosquitoes collected from the UFFs, even during recent epidemics in the city of Salinas,[19] suggests that these viruses were not actively circulating in these environments. Sylvatic cycles of DENV and ZIKV persist in Asia and Africa, where both viruses continue to spillover to humans and are occasionally translocated to new continents.[66,67,68] Furthermore, reports in the literature describe natural infection of the sylvatic mosquito *Haemagogus leucocelaenus* in Brazil,[69] raising concerns about the potential establishment of a sylvatic dengue cycle. Therefore, continuous entomo-virological surveillance is essential to monitor the risk of sylvatic transmission cycles of DENV and ZIKV becoming established in the Americas.[70,71] Although the risk of spillover or spillback involves a complex interplay of factors,[72] the presence of known vector species identified in this study highlights a concerning issue for public health authorities.

Finally, it is important to acknowledge some limitations of our study. Mosquito collections were conducted during an active dengue outbreak and over a restricted temporal window (February to June 2024). These conditions may have influenced the observed species composition, infection rates, and viral circulation patterns, thereby limiting the generalisability of our findings to other time periods or non-outbreak contexts.

*In conclusion* - Our study underscores the critical importance of investigating mosquito fauna in conjunction with virological analysis in urban and peri-urban forested areas, which are often excluded from municipal control and surveillance programs. Despite the limitations of our study, such as the short sampling period, our findings contribute to understanding the role of mosquito fauna in the epidemiology of DENV and provide valuable insights into the dynamics of mosquito assemblages in urban green areas. Collectively, these data can inform the development of strategies to assess the receptivity of different areas to arboviruses and guide the design of preventive measures aimed at managing the transition between urban and forest environments, with a focus on species that may serve as bridge vectors. Integrating entomological and virological surveillance across domiciliary and green areas strengthens early warning systems by detecting arboviruses in field-caught vectors and enabling timely control measures. Furthermore, the molecular monitoring approach can be incorporated into routine entomological programs to enhance preparedness and support evidence-based outbreak prevention strategies.

### ACKNOWLEDGEMENTS

To the Salinas City Hall and all Endemic Disease Control Agents (*Agentes de Combate às Endemias*) for their valuable support during fieldwork, and the Ezequiel Dias Foundation (FUNED) for providing the infrastructure for this Laboratory and Genomic Surveillance project.

### AUTHORS' CONTRIBUTION

CHO and FVSA - conceptualisation; ABA, ATF, CHO, FVSA, TJT and WSA - data curation; CHO, FCMI, LMRT, NRG, TERA, RMCB, TJT and WSA - formal analysis; DCCC, FCMI, FSC, LCJA and FVSA - funding acquisition; ABA, ATF CHO, NRG, RMCB, TERA and TJT - investigation; ABA, ATF CHO, LMRT, NRG, RMCB, TERA, TJT and FVSA - methodology; CHO, FSC, FVSA, LMRT and LCJA - project administration; DCCC, FCMI, FSC, FVSA and LCJA - resources; FSC, FVSA and LCJA - supervision; CHO, FCMI, FSC and FVSA, NRG, TERA - validation; ABA, ATF, CHO,

FVSA, LMRT, RMCB and TJT - visualisation; CHO, FCMI, FSC, FVSA, LMRT, WSA - writing-original draft preparation; ABA, ATF, DCCC, LCJA, NRG, RMCB, TERA and TJT - writing-review & editing preparation. All authors have read and agreed to the published version of the manuscript. The authors declare no conflict of interest. The funders had no role in study design, data collection and analysis, decision to publish, or preparation of this manuscript.

## DATA AVAILABILITY

The authors declare that all data supporting the findings of this study are available within the paper. The database used in this work for phylogenetic analysis was created using DENV-1 genomic sequences deposited in the GISAID EpiArbo database (GISAID Identifier: EPI_SET_250212gv; doi: 10.55876/gis8.250212gv). We sincerely thank all the laboratories and researchers who generated and shared these sequences, enabling advances in genomic surveillance and research. For details about the contributors, including accession numbers, virus names, collection dates, originating and submitting laboratories, and author lists, please visit 10.55876/gis8.250212gv. The DENV-1 genome generated in this study has been deposited in the GISAID EpiArbo database under the following Accession ID: EPI_ISL_19725409.

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

# OPEN PEER REVIEW

Memórias do IOC thanks the anonymous reviewers for their contribution to the peer review of this work.

## FIRST REVIEW ROUND

REVIEWERS' COMMENTS

### REVIEWER #1

General Comments

The study presents interesting and valuable data on arbovirus surveillance and vector control, using molecular and/or entomological approaches. The manuscript is generally clear and well-organized. However, I suggest the following revisions to improve the clarity, reproducibility, and impact of the paper.

Major Comments

- The introduction could more clearly articulate the novelty of the study. Please highlight what is unique about your approach or findings compared to similar studies from the region or previous work by your group.

• Methodology aspects

- Please provide more detailed descriptions for critical steps, such as:

- Although collection methodology is previously described, please inform what type of trap was used, if the study has an ethical committee number approval…

- Exact RNA input amounts for cDNA synthesis and qPCR reactions.

- Criteria used to define positive infection or transmission in qPCR assays. Why did authors choose to use liquid nitrogen, instead of deep freezing samples? This strategy to acclimate samples from -196 to -20 can result in fragmentation of RNA genomes;

- Strain origin and viral titers used for infections, if applicable.

- Include catalog numbers for commercial kits or reagents where relevant.

• Results

- In figures presenting infection or dissemination rates, consider including statistical annotations (e.g., p-values or confidence intervals).

- Define all abbreviations used in the figure legends.

- Indicate how replicates were handled (biological vs. technical) and mention any exclusions.

• Discussion and Interpretation

- Expand the comparison with findings from other geographic regions or vector populations. Results are compared with relevant national and international literature. However, some recent publications from 2023–2024 could be more thoroughly discussed to reinforce the novelty and implications of the findings.

- The study would benefit from a more explicit discussion on potential biases, particularly regarding the sampling duration and spatial coverage, which may influence species richness and virus detection rates.

- was there any report of DENV-1 in humans from that region?

- The conclusion could more explicitly highlight how the findings can inform future control strategies or surveillance efforts.

Minor Comments

- Abstract: Specify whether the study is observational, experimental, or integrative.

- Figures: Please ensure all figures are of sufficient resolution for publication. Some appear pixelated in the current version.

- Several minor typos are present (e.g., 'greater' should be 'higher' in scientific context when referring to abundance).

* Line 296: 'is higher to those' should be corrected to 'is higher than those'.

* Line 315: 'eclectic in their blood-feeding habits' – consider rephrasing to 'eclectic blood-feeding behavior'.

- Consider standardizing genus/species names in italics throughout the manuscript.

- Some acronyms such as UFF should be defined at first mention in both abstract and main text.

- References: Consider including recent literature (post-2022) to reflect current studies in arbovirus-vector research.

### REVIEWER #2

Outbreak in Salinas, MG, Brazil

Summary and General Assessment:

This manuscript presents an important and timely investigation into mosquito fauna and dengue virus circulation in urban and peri-urban environments during a dengue outbreak in Salinas, Minas Gerais, Brazil. The study is well-designed, combining entomological, virological, ecological, and genomic data. The integration of field collec-

tions, molecular detection of arboviruses, and phylogenomic analysis of DENV-1 provides a comprehensive approach. The conclusions are, in general, well supported by the results. The manuscript contributes significantly to the field of entomo-virology and public health surveillance, and I recommend it for publication after minor revisions.

Minor Comments and Suggestions for Improvement:

1. Line 28 – "Many" sounds too strong

Suggest changing "Many species exhibit high levels of anthropophily…" to "Several species exhibit high levels of anthropophily…" as we do not have evidence for a large number of species with this trait.

2. Line 137 – Dissection procedure

A brief description of the head separation process should be included, especially to clarify measures taken to avoid cross-contamination during dissection.

3. Line 138 – RNA extraction method

Although the authors refer to a previous study for the methodology, a brief summary of the RNA extraction procedure would help the reader and enhance reproducibility.

4. Line 214 – Clarify the procedure for positive pools

Please clarify that for pools that tested positive for dengue, RNA was extracted individually from mosquito heads and tested separately to confirm viral dissemination.

5. Line 237 – Figure font size

The font size in Figure 2 (phylogenetic tree) is quite small. Increasing the font size would improve readability, especially for the clade labels.

6. Line 273 – Verb tense in the Discussion

Although not incorrect, the use of the simple past tense in the Discussion is not ideal. The authors are encouraged to consider using the present tense when interpreting their findings, as is more standard in scientific discussions.

7. Line 275 – Wording suggestion for caution

The phrase "with no other species implicated in the transmission" could be softened to "with no evidence of other species implicated in the transmission," to reflect limitations of the detection methods.

8. Line 297 – Minimum infection rate (MIR) comparison

The values from the literature cited for MIR (e.g., 0.35 for Ae. albopictus and 3.37 for Ae. aegypti) should be more clearly contextualized, including the geographic and temporal setting of those previous studies for a more appropriate comparison.

9. Line 298 – Study limitations and context

The discussion would benefit from a clearer framing of the characteristics and limitations of the study. Specifically, emphasize that the mosquito collections were conducted during an outbreak period and over a limited temporal window, which may influence the representativeness of the findings.

Recommendation:

Minor Revisions

This manuscript is scientifically sound and relevant. I recommend it for publication pending minor revisions addressing the points above.

## AUTHORS' RESPONSE TO THE REVIEWERS

Review response MIOC-2025-0086

Salinas, August 27, 2025

Dear Editor,

We sincerely appreciate the time and effort the reviewers have dedicated to evaluating our manuscript, "Entomo-virological investigation in urban forest fragments and intradomiciles during a Dengue outbreak in Salinas, MG, Brazil". Their valuable feedback has been instrumental in refining and strengthening our work.

All the reviewers' suggestions and comments have been carefully considered and incorporated into the revised manuscript. In addition, we have prepared a detailed response letter addressing each point raised by the reviewers. We believe these revisions have significantly improved the clarity, accuracy, and overall quality of the manuscript.

We thank you for the opportunity to revise our work, and we look forward to your consideration for publication.

Best regards,

Cirilo H. Oliveira and Filipe V. S. Abreu

Reviewer #1

General Comments

The study presents interesting and valuable data on arbovirus surveillance and vector control, using molecular and/or entomological approaches. The manuscript is generally clear and well-organized. However, I suggest the following revisions to improve the clarity, reproducibility, and impact of the paper.

Major Comments

- The introduction could more clearly articulate the novelty of the study. Please highlight what is unique about your approach or findings compared to similar studies from the region or previous work by your group.

R.: We appreciate the reviewer's insightful comment. In the revised version of the manuscript, we have clari-

fied the novelty of our study in the final paragraph of the Introduction section. Specifically, we emphasize that this is the first entomo-virological investigation integrating molecular detection and ecological analysis of mosquito assemblages in urban forest fragments and intradomiciles during an active dengue outbreak in the city of Salinas, northern Minas Gerais. Our study also generated the first complete DENV-1 genome sequence from an *Aedes aegypti* population in this region, which we analyzed using phylogenomic tools. Please, see lines 82-86.

• Methodology aspects

- Please provide more detailed descriptions for critical steps, such as:

- Although collection methodology is previously described, please inform what type of trap was used, if the study has an ethical committee number approval.

R.: We thank the reviewer for this observation. In response, we have revised the Materials and Methods section to include more detailed descriptions of the mosquito collection. Specifically:

We clarified that in intradomiciles, mosquitoes were collected using battery-powered Nasci aspirators. We added the information: "Briefly, household visits were scheduled according to residents' availability and conducted by a municipal vector surveillance agent and an entomologist equipped with battery-powered Nasci aspirators, oral aspirators, and entomological cages. Sampling involved thorough inspection of all rooms, with special attention to hidden niches such as under furniture and behind cabinets." Please, see lines 103-107.

Regarding ethical aspects, we have included the authorization number for mosquito collection issued by the Brazilian environmental authorities, as follows: "Methods and protocols were previously approved by Brazilian Ministry of the Environment (SISBIO nº 75826-4)." (Please, see lines 100-102). Approval from an animal or human research ethics committee was not required, as the field activities were carried out in collaboration with the Municipal Health Department of Salinas, in accordance with institutional agreements for surveillance activities.

- Exact RNA input amounts for cDNA synthesis and qPCR reactions.

R.: We appreciate the reviewer's suggestion. In the revised version of the manuscript, we added this information: "RNA from body pools was tested for DENV, ZIKV, and CHIKV by RT-qPCR using the Bioclin ZDC Multiplex One-Step kit, with 9 µL (~1µg) of total RNA as input, following the manufacturer's protocol." Please, see lines 158, 162-163.

- Criteria used to define positive infection or transmission in qPCR assays. Why did authors choose to use liquid nitrogen, instead of deep-freezing samples? This strategy to acclimate samples from -196 to -20 can result in fragmentation of RNA genomes;

R.: We thank the reviewer for raising these important methodological points. Regarding the criteria used to define positive infection, we considered a mosquito pool or head as positive when the cycle threshold (Ct) value was below 40. We now added this information. Please, see lines 162-163.

Regarding the storage strategy, liquid nitrogen (-196 °C) was employed due to the limited availability of -80 °C freezers in our facilities. To mitigate potential RNA fragmentation during temperature transition, mosquitoes were processed immediately after removal from liquid nitrogen, preventing additional freeze–thaw cycles. This procedure appears to have preserved viral RNA integrity, as evidenced by the high genome coverage and sequencing depth obtained. Nevertheless, we recognize the reviewer's concern and will aim to use -80 °C freezers in future experiments whenever feasible.

- Strain origin and viral titers used for infections, if applicable.

R.: We thank the reviewer for this observation. We would like to clarify that this study focused exclusively on the molecular detection of arboviruses in field-collected mosquitoes. No experimental infections were performed; therefore, no viral strains or titers were used.

- Include catalog numbers for commercial kits or reagents where relevant.

R.: Thank you for the suggestion. We have now included catalog numbers for all commercial kits and reagents mentioned in the Material and Methods section to enhance reproducibility. Please, see the Material and Methods section.

• Results

- In figures presenting infection or dissemination rates, consider including statistical annotations (e.g., p-values or confidence intervals).

R.: We thank the reviewer for this suggestion. However, no statistical analyses or figures presenting infection or dissemination rates were performed, as these assessments were not within the scope or objectives of the present study. Nevertheless, we acknowledge the importance of such analyses and will consider including them in future investigations.

- Define all abbreviations used in the figure legends.

R.: Thank you for pointing this out. We have now revised all figure legends and tables to ensure that all abbreviations are clearly defined at their first mention. The changes can be found in the revised legends for Figure 4 and Table 1.

- Indicate how replicates were handled (biological vs. technical) and mention any exclusions.

R.: Thank you for this important observation. We have clarified in the revised Materials and Methods section that qPCR reaction was performed in technical duplicates to ensure reproducibility (Please, see line 162). No data points were excluded from the analysis.

• Discussion and Interpretation

- Expand the comparison with findings from other geographic regions or vector populations.

R.: Thank you for this suggestion. We have expanded the Discussion to include other studies. These additions can be found in lines 292-296 and 340-350 of the revised manuscript. The following references were added:

28. Edillo F, Ymbong RR, Navarro AO, Cabahug MM, Saavedra K. Detecting the impacts of humidity, rainfall, temperature, and season on chikungunya, dengue and Zika viruses in Aedes albopictus mosquitoes from selected sites in Cebu city, Philippines. Virol J. 2024 Feb 15;21(1):42.

29. Dissanayake DS, Wijekoon CD, Wegiriya H. Diversity of mosquito natural enemies and their feeding efficacy on Aedes vectors. J Vector Borne Dis. 2024 Oct;61(4):564–73.

30. Liu Q, Wang J, Hou J, Wu Y, Zhang H, Xing D, et al. Entomological Investigation and Detection of Dengue Virus Type 1 in Aedes (Stegomyia) albopictus (Skuse) During the 2018–2020 Outbreak in Zhejiang Province, China. Front Cell Infect Microbiol. 2022 Jul 1;12.

31. Gomes EO, Sacchetto L, Teixeira M, Chaves BA, Hendy A, Mendonça C, et al. Detection of Zika Virus in Aedes aegypti and Aedes albopictus Mosquitoes Collected in Urban Forest Fragments in the Brazilian Amazon. Viruses. 2023 Jun 12;15(6):1356.

32. Bedoya-Rodríguez FJ, Guevara-Fletcher CE, Pelegrin-Ramírez JS. Diversity analysis, distribution and abundance of mosquito (Diptera: Culicidae) assemblages at urban sector from southwestern Colombia. Biologia (Bratisl). 2025 Jan 16;80(3):561–72.

33. Ataide LMS, Moura CA, Fernandes JC, Araújo AB, Izumisawa CM, Figueiredo JVA, et al. Species composition and abundance of mosquitoes (Diptera: Culicidae) in a green area surrounded by urbanization in the Neotropical megacity São Paulo, Brazil. Entomological Communications. 2023 Dec 20;5:ec05041.

34. Borges MAZ, Silva HR da, Rodrigues CP, Santos CF, Souza GA de, Junior RR, et al. Biodiversidade de mosquitos (Diptera: Culicidae) em parques urbanos de Montes Claros – MG. Revista Unimontes Científica. 2024 Oct 29;26(2).

35. Heinisch MR, Medeiros-Sousa AR, Andrade PS, Urbinatti PR, Almeida RMMS, Lima-Camara TN. Fauna and Virological Investigation of Mosquitoes in Urban Parks in São Paulo, Brazil. J Am Mosq Control Assoc. 2023 Jun;39(2):75–84.

54. Hendy A, Hernandez-Acosta E, Valério D, Fé NF, Mendonça CR, Costa ER, et al. Where boundaries become bridges: Mosquito community composition, key vectors, and environmental associations at forest edges in the central Brazilian Amazon. PLoS Negl Trop Dis. 2023 Apr 26;17(4):e0011296.

55. Santos TP. Potential of Aedes albopictus as a bridge vector for enzootic pathogens at the urban-forest interfacein Brazil. Emerg Microbes Infect. 2018 Oct;

- Results are compared with relevant national and international literature. However, some recent publications from 2023–2024 could be more thoroughly discussed to reinforce the novelty and implications of the findings.

R.: Thank you for this valuable comment. As previously stated, we have revised the Discussion section to relevant studies published in 2023 and 2024.

- The study would benefit from a more explicit discussion on potential biases, particularly regarding the sampling duration and spatial coverage, which may influence species richness and virus detection rates.

R.: Thank you for your thoughtful comment. We have revised the Discussion section to explicitly address potential limitations of our study related to sampling duration and spatial coverage. The fllowing paragraph was added "Finally, it is important to acknowledge some limitations of our study. Mosquito collections were conducted during an active dengue outbreak and over a restricted temporal window (February to March 2024). These conditions may have influenced the observed species composition, infection rates, and viral circulation patterns, thereby limiting the generalizability of our findings to other time periods or non-outbreak contexts.". Please, see lines 382-386.

- was there any report of DENV-1 in humans from that region?

R.: Yes, DENV-1 circulation in humans has been previously reported in the study region. We have now made this information clearer in the revised Discussion (lines 306–307) to contextualize our findings and support the relevance of vector-based viral detection.

- The conclusion could more explicitly highlight how the findings can inform future control strategies or surveillance efforts.

R.: Thank you for the suggestion. We have revised the Conclusion section to more clearly emphasize the practical implications of our findings. Please, see lines 396-401.

Minor Comments

- Abstract: Specify whether the study is observational, experimental, or integrative.

R.: Thank you for the observation. We have revised the abstract to explicitly state that this is an integrative study combining mosquito collection, arbovirus detection and sequencing, and ecological analyses. This clarification appears in the revised abstract (lines 29–30).

- Figures: Please ensure all figures are of sufficient resolution for publication. Some appear pixelated in the current version.

R.: Thank you for this comment. We have reviewed all figures; however, the submission process compresses their quality. We will provide high-resolution versions of all figures separately.

Several minor typos are present (e.g., 'greater' should be 'higher' in scientific context when referring to abundance).
* Line 296: 'is higher to those' should be corrected to 'is higher than those'.
* Line 315: 'eclectic in their blood-feeding habits' – consider rephrasing to 'eclectic blood-feeding behavior'.
R.: Thank you for your careful reading and helpful suggestions. "Greater" was replaced by "higher" through the manuscript;
Line 296 (now, 318) was corrected to "is higher than those";
Line 315 (now, 336-337) was rephrased to "eclectic blood-feeding behavior" to improve clarity and style.
- Consider standardizing genus/species names in italics throughout the manuscript.
R.: Thank you for the suggestion. We carefully revised the entire manuscript to ensure that all genus and species names are consistently formatted in italics, in accordance with scientific conventions.
- Some acronyms such as UFF should be defined at first mention in both abstract and main text.
R.: Thank you for the observation. We have reviewed the manuscript and ensured that all acronyms, including UFF (Urban Forest Fragment), are now defined at their first mention in both the abstract and the main text.
- References: Consider including recent literature (post-2022) to reflect current studies in arbovirus-vector research.
R.: Thank you for this valuable suggestion. We have updated the reference list to include eight recent publications (from 2023 to 2025) that are relevant to arbovirus-vector research.

Reviewer #2
Summary and General Assessment:
This manuscript presents an important and timely investigation into mosquito fauna and dengue virus circulation in urban and peri-urban environments during a dengue outbreak in Salinas, Minas Gerais, Brazil. The study is well-designed, combining entomological, virological, ecological, and genomic data. The integration of field collections, molecular detection of arboviruses, and phylogenomic analysis of DENV-1 provides a comprehensive approach. The conclusions are, in general, well supported by the results. The manuscript contributes significantly to the field of entomo-virology and public health surveillance, and I recommend it for publication after minor revisions.
R.: We sincerely thank the reviewer for the positive and encouraging assessment of our manuscript. We are glad that the study's multidisciplinary approach and its contributions to entomo-virology and public health surveillance were appreciated. We have addressed the minor revisions suggested in the specific comments below to further improve the clarity and quality of the manuscript. Thank you again for your thoughtful review and recommendation for publication.
Minor Comments and Suggestions for Improvement:
1. Line 28 – "Many" sounds too strong
Suggest changing "Many species exhibit high levels of anthropophily…" to "Several species exhibit high levels of anthropophily…" as we do not have evidence for a large number of species with this trait.
R.: Thank you for the suggestion. We agree that "Several" is more appropriate in this context. The sentence has been revised to: "Several species exhibit high levels of anthropophily…" Please, see line 28.
2. Line 137 – Dissection procedure
A brief description of the head separation process should be included, especially to clarify measures taken to avoid cross-contamination during dissection.
R.: Thank you for this important observation. We have added a brief description of the dissection procedure to clarify the methodology and precautions taken to avoid cross-contamination. The revised sentence now reads: "Each mosquito's head was carefully separated from its body under a stereomicroscope, using a sterile scalpel blade dedicated to that specimen to prevent cross-contamination.". Please, see lines 147-149.
3. Line 138 – RNA extraction method
Although the authors refer to a previous study for the methodology, a brief summary of the RNA extraction procedure would help the reader and enhance reproducibility.
R.: Thank you for the suggestion. We have now added the following sentences to the manuscript: "Briefly, individual heads and pools of up to 10 bodies from the same species and sex were placed in enriched L-15 medium (20% fetal bovine serum, 0.5% non-essential amino acids, 1% penicillin, 0.1% gentamicin, and 0.1% fungizone). Tissues were homogenized with beads in a beadbeater for 30 s at 7500 rpm, followed by centrifugation at 12,000 rpm for 8 min at 4 °C. A 140 μL aliquot of the resulting supernatant was used for RNA extraction with the QIAamp Viral RNA Mini Kit (Qiagen), according to the manufacturer's instructions.". Please, see lines 150-156.
4. Line 214 – Clarify the procedure for positive pools
Please clarify that for pools that tested positive for dengue, RNA was extracted individually from mosquito heads and tested separately to confirm viral dissemination.
R.: Thank you for the observation. We have clarified this point in the revised manuscript. The sentence now reads: "For DENV-positive pools, RNA from each mosquito head was individually tested to confirm viral dissemination". Please, see lines 159-160.

5. Line 237 – Figure font size

The font size in Figure 2 (phylogenetic tree) is quite small. Increasing the font size would improve readability, especially for the clade labels.

R.: Thank you for the suggestion. We have increased the font size of the labels in Figure 2 to the maximum possible, given space limitations. Please, see Figure 2 in the new manuscript version.

6. Line 273 – Verb tense in the Discussion

Although not incorrect, the use of the simple past tense in the Discussion is not ideal. The authors are encouraged to consider using the present tense when interpreting their findings, as is more standard in scientific discussions.

R.: Thank you for the observation. We have revised the Discussion section to use the present tense where appropriate.

7. Line 275 – Wording suggestion for caution

The phrase "with no other species implicated in the transmission" could be softened to "with no evidence of other species implicated in the transmission," to reflect limitations of the detection methods.

R.: Thank you for the suggestion. We have adjusted the sentence to: "with no evidence of other species implicated in the transmission," in order to more accurately reflect the limitations of our detection methods and avoid overstatement. Please, see line: 291.

8. Line 297 – Minimum infection rate (MIR) comparison

The values from the literature cited for MIR (e.g., 0.35 for Ae. albopictus and 3.37 for Ae. aegypti) should be more clearly contextualized, including the geographic and temporal setting of those previous studies for a more appropriate comparison.

R.: Thank you for this valuable observation. We have revised the text to provide clearer context for the MIR values cited. Specifically, we now state: "The minimum infection rate (MIR) for Ae. aegypti observed in this study is higher than those previously reported in the literature (MIR = 1.47 for adult Ae. albopictus collected in spring 2014 in São Paulo, Brazil, and 3.37 and 16.2 for immature and adult Ae. aegypti, respectively, collected in Rio Grande do Norte, Brazil, between 2011 and 2014).". Please, see lines 317-321.

9. Line 298 – Study limitations and context

The discussion would benefit from a clearer framing of the characteristics and limitations of the study. Specifically, emphasize that the mosquito collections were conducted during an outbreak period and over a limited temporal window, which may influence the representativeness of the findings.

R.: Thank you for the suggestion. We have revised the discussion to include a clearer statement of the study's limitations. The following paragraph was added: "Finally, it is important to acknowledge some limitations of our study. Mosquito collections were conducted during an active dengue outbreak and over a restricted temporal window (February to March 2024). These conditions may have influenced the observed species composition, infection rates, and viral circulation patterns, thereby limiting the generalizability of our findings to other time periods or non-outbreak contexts.". Please, see lines 382-386.

Recommendation:

Minor Revisions

This manuscript is scientifically sound and relevant. I recommend it for publication pending minor revisions addressing the points above.

R.: We sincerely thank the reviewer for the positive evaluation and thoughtful comments that helped us improve the clarity, rigor, and impact of our manuscript. All suggested revisions were carefully addressed point by point, and the text was revised accordingly. We believe the updated version now meets the expectations for publication and we remain available for any further clarifications.

## SECOND REVIEW ROUND

### REVIEWERS' COMMENTS

### REVIEWER #1

The authors have carefully addressed all the comments and suggestions raised in my previous review. The revised version of the manuscript has been significantly improved, and in my opinion, it now meets the requirements for publication. I recommend acceptance of the article in its current form.

### REVIEWER #2

The authors have addressed and the concerns from my last review and the paper, now, is ready for publication.

