## [Reviewer Report · FIRST REVIEW ROUND - REVIEWERS COMMENTS]

## REVIEWER #1

General Comments

The study presents interesting and valuable data on arbovirus surveillance and vector control, using molecular and/or entomological approaches. The manuscript is generally clear and well-organized. However, I suggest the following revisions to improve the clarity, reproducibility, and impact of the paper.

Major Comments

- The introduction could more clearly articulate the novelty of the study. Please highlight what is unique about your approach or findings compared to similar studies from the region or previous work by your group.

• Methodology aspects

- Please provide more detailed descriptions for critical steps, such as:

- Although collection methodology is previously described, please inform what type of trap was used, if the study has an ethical committee number approval…

- Exact RNA input amounts for cDNA synthesis and qPCR reactions.

- Criteria used to define positive infection or transmission in qPCR assays. Why did authors choose to use liquid nitrogen, instead of deep freezing samples? This strategy to acclimate samples from -196 to -20 can result in fragmentation of RNA genomes;

- Strain origin and viral titers used for infections, if applicable.

- Include catalog numbers for commercial kits or reagents where relevant.

• Results

- In figures presenting infection or dissemination rates, consider including statistical annotations (e.g., p-values or confidence intervals).

- Define all abbreviations used in the figure legends.

- Indicate how replicates were handled (biological vs. technical) and mention any exclusions.

• Discussion and Interpretation

- Expand the comparison with findings from other geographic regions or vector populations.

Results are compared with relevant national and international literature. However, some recent publications from 2023–2024 could be more thoroughly discussed to reinforce the novelty and implications of the findings.

- The study would benefit from a more explicit discussion on potential biases, particularly regarding the sampling duration and spatial coverage, which may influence species richness and virus detection rates.

- was there any report of DENV-1 in humans from that region?

- The conclusion could more explicitly highlight how the findings can inform future control strategies or surveillance efforts.

Minor Comments

- Abstract: Specify whether the study is observational, experimental, or integrative.

- Figures: Please ensure all figures are of sufficient resolution for publication. Some appear pixelated in the current version.

- Several minor typos are present (e.g., ‘greater’ should be ‘higher’ in scientific context when referring to abundance).

* Line 296: ‘is higher to those’ should be corrected to ‘is higher than those’.

* Line 315: ‘eclectic in their blood-feeding habits’ – consider rephrasing to ‘eclectic blood-feeding behavior’.

- Consider standardizing genus/species names in italics throughout the manuscript.

- Some acronyms such as UFF should be defined at first mention in both abstract and main text.

- References: Consider including recent literature (post-2022) to reflect current studies in arbovirus-vector research.

## REVIEWER #2

Outbreak in Salinas, MG, Brazil

Summary and General Assessment:

This manuscript presents an important and timely investigation into mosquito fauna and dengue virus circulation in urban and peri-urban environments during a dengue outbreak in Salinas, Minas Gerais, Brazil. The study is well-designed, combining entomological, virological, ecological, and genomic data. The integration of field collections, molecular detection of arboviruses, and phylogenomic analysis of DENV-1 provides a comprehensive approach. The conclusions are, in general, well supported by the results. The manuscript contributes significantly to the field of entomo-virology and public health surveillance, and I recommend it for publication after minor revisions.

Minor Comments and Suggestions for Improvement:

1. Line 28 – “Many” sounds too strong

Suggest changing “Many species exhibit high levels of anthropophily…” to “Several species exhibit high levels of anthropophily…” as we do not have evidence for a large number of species with this trait.

2. Line 137 – Dissection procedure

A brief description of the head separation process should be included, especially to clarify measures taken to avoid cross-contamination during dissection.

3. Line 138 – RNA extraction method

Although the authors refer to a previous study for the methodology, a brief summary of the RNA extraction procedure would help the reader and enhance reproducibility.

4. Line 214 – Clarify the procedure for positive pools

Please clarify that for pools that tested positive for dengue, RNA was extracted individually from mosquito heads and tested separately to confirm viral dissemination.

5. Line 237 – Figure font size

The font size in Figure 2 (phylogenetic tree) is quite small. Increasing the font size would improve readability, especially for the clade labels.

6. Line 273 – Verb tense in the Discussion

Although not incorrect, the use of the simple past tense in the Discussion is not ideal. The authors are encouraged to consider using the present tense when interpreting their findings, as is more standard in scientific discussions.

7. Line 275 – Wording suggestion for caution

The phrase “with no other species implicated in the transmission” could be softened to “with no evidence of other species implicated in the transmission,” to reflect limitations of the detection methods.

8. Line 297 – Minimum infection rate (MIR) comparison

The values from the literature cited for MIR (e.g., 0.35 for Ae. albopictus and 3.37 for Ae. aegypti) should be more clearly contextualized, including the geographic and temporal setting of those previous studies for a more appropriate comparison.

9. Line 298 – Study limitations and context

The discussion would benefit from a clearer framing of the characteristics and limitations of the study. Specifically, emphasize that the mosquito collections were conducted during an outbreak period and over a limited temporal window, which may influence the representativeness of the findings.

Recommendation:

Minor Revisions

This manuscript is scientifically sound and relevant. I recommend it for publication pending minor revisions addressing the points above.

---

## [Author Response · AUTHORS RESPONSE TO REVIEWERS]

## Review response MIOC-2025-0086

Salinas, August 27, 2025

Dear Editor,

We sincerely appreciate the time and effort the reviewers have dedicated to evaluating our manuscript, “Entomo-virological investigation in urban forest fragments and intradomiciles during a Dengue outbreak in Salinas, MG, Brazil”. Their valuable feedback has been instrumental in refining and strengthening our work.

All the reviewers’ suggestions and comments have been carefully considered and incorporated into the revised manuscript. In addition, we have prepared a detailed response letter addressing each point raised by the reviewers. We believe these revisions have significantly improved the clarity, accuracy, and overall quality of the manuscript.

We thank you for the opportunity to revise our work, and we look forward to your consideration for publication.

Best regards,

Cirilo H. Oliveira and Filipe V. S. Abreu

## Reviewer #1

General Comments

The study presents interesting and valuable data on arbovirus surveillance and vector control, using molecular and/or entomological approaches. The manuscript is generally clear and well-organized. However, I suggest the following revisions to improve the clarity, reproducibility, and impact of the paper.

Major Comments

- The introduction could more clearly articulate the novelty of the study. Please highlight what is unique about your approach or findings compared to similar studies from the region or previous work by your group.

R.: We appreciate the reviewer’s insightful comment. In the revised version of the manuscript, we have clarified the novelty of our study in the final paragraph of the Introduction section. Specifically, we emphasize that this is the first entomo-virological investigation integrating molecular detection and ecological analysis of mosquito assemblages in urban forest fragments and intradomiciles during an active dengue outbreak in the city of Salinas, northern Minas Gerais. Our study also generated the first complete DENV-1 genome sequence from an Aedes aegypti population in this region, which we analyzed using phylogenomic tools. Please, see lines 82-86.

• Methodology aspects

- Please provide more detailed descriptions for critical steps, such as:

- Although collection methodology is previously described, please inform what type of trap was used, if the study has an ethical committee number approval.

R.: We thank the reviewer for this observation. In response, we have revised the Materials and Methods section to include more detailed descriptions of the mosquito collection. Specifically:

We clarified that in intradomiciles, mosquitoes were collected using battery-powered Nasci aspirators. We added the information: “Briefly, household visits were scheduled according to residents’ availability and conducted by a municipal vector surveillance agent and an entomologist equipped with battery-powered Nasci aspirators, oral aspirators, and entomological cages. Sampling involved thorough inspection of all rooms, with special attention to hidden niches such as under furniture and behind cabinets.” Please, see lines 103-107.

Regarding ethical aspects, we have included the authorization number for mosquito collection issued by the Brazilian environmental authorities, as follows: “Methods and protocols were previously approved by Brazilian Ministry of the Environment (SISBIO nº 75826-4).” (Please, see lines 100-102). Approval from an animal or human research ethics committee was not required, as the field activities were carried out in collaboration with the Municipal Health Department of Salinas, in accordance with institutional agreements for surveillance activities.

- Exact RNA input amounts for cDNA synthesis and qPCR reactions.

R.: We appreciate the reviewer’s suggestion. In the revised version of the manuscript, we added this information: “RNA from body pools was tested for DENV, ZIKV, and CHIKV by RT-qPCR using the Bioclin ZDC Multiplex One-Step kit, with 9 µL (~1µg) of total RNA as input, following the manufacturer’s protocol.” Please, see lines 158, 162-163.

- Criteria used to define positive infection or transmission in qPCR assays. Why did authors choose to use liquid nitrogen, instead of deep-freezing samples? This strategy to acclimate samples from -196 to -20 can result in fragmentation of RNA genomes;

R.: We thank the reviewer for raising these important methodological points. Regarding the criteria used to define positive infection, we considered a mosquito pool or head as positive when the cycle threshold (Ct) value was below 40. We now added this information. Please, see lines 162-163.

Regarding the storage strategy, liquid nitrogen (-196 °C) was employed due to the limited availability of -80 °C freezers in our facilities. To mitigate potential RNA fragmentation during temperature transition, mosquitoes were processed immediately after removal from liquid nitrogen, preventing additional freeze–thaw cycles. This procedure appears to have preserved viral RNA integrity, as evidenced by the high genome coverage and sequencing depth obtained. Nevertheless, we recognize the reviewer’s concern and will aim to use -80 °C freezers in future experiments whenever feasible.

- Strain origin and viral titers used for infections, if applicable.

R.: We thank the reviewer for this observation. We would like to clarify that this study focused exclusively on the molecular detection of arboviruses in field-collected mosquitoes. No experimental infections were performed; therefore, no viral strains or titers were used.

- Include catalog numbers for commercial kits or reagents where relevant.

R.: Thank you for the suggestion. We have now included catalog numbers for all commercial kits and reagents mentioned in the Material and Methods section to enhance reproducibility. Please, see the Material and Methods section.

• Results

- In figures presenting infection or dissemination rates, consider including statistical annotations (e.g., p-values or confidence intervals).

R.: We thank the reviewer for this suggestion. However, no statistical analyses or figures presenting infection or dissemination rates were performed, as these assessments were not within the scope or objectives of the present study. Nevertheless, we acknowledge the importance of such analyses and will consider including them in future investigations.

- Define all abbreviations used in the figure legends.

R.: Thank you for pointing this out. We have now revised all figure legends and tables to ensure that all abbreviations are clearly defined at their first mention. The changes can be found in the revised legends for Figure 4 and Table 1.

- Indicate how replicates were handled (biological vs. technical) and mention any exclusions.

R.: Thank you for this important observation. We have clarified in the revised Materials and Methods section that qPCR reaction was performed in technical duplicates to ensure reproducibility (Please, see line 162). No data points were excluded from the analysis.

• Discussion and Interpretation

- Expand the comparison with findings from other geographic regions or vector populations.

R.: Thank you for this suggestion. We have expanded the Discussion to include other studies. These additions can be found in lines 292-296 and 340-350 of the revised manuscript. The following references were added:

28. Edillo F, Ymbong RR, Navarro AO, Cabahug MM, Saavedra K. Detecting the impacts of humidity, rainfall, temperature, and season on chikungunya, dengue and Zika viruses in Aedes albopictus mosquitoes from selected sites in Cebu city, Philippines. Virol J. 2024 Feb 15;21(1):42.

29. Dissanayake DS, Wijekoon CD, Wegiriya H. Diversity of mosquito natural enemies and their feeding efficacy on Aedes vectors. J Vector Borne Dis. 2024 Oct;61(4):564–73.

30. Liu Q, Wang J, Hou J, Wu Y, Zhang H, Xing D, et al. Entomological Investigation and Detection of Dengue Virus Type 1 in Aedes (Stegomyia) albopictus (Skuse) During the 2018–2020 Outbreak in Zhejiang Province, China. Front Cell Infect Microbiol. 2022 Jul 1;12.

31. Gomes EO, Sacchetto L, Teixeira M, Chaves BA, Hendy A, Mendonça C, et al. Detection of Zika Virus in Aedes aegypti and Aedes albopictus Mosquitoes Collected in Urban Forest Fragments in the Brazilian Amazon. Viruses. 2023 Jun 12;15(6):1356.

32. Bedoya-Rodríguez FJ, Guevara-Fletcher CE, Pelegrin-Ramírez JS. Diversity analysis, distribution and abundance of mosquito (Diptera: Culicidae) assemblages at urban sector from southwestern Colombia. Biologia (Bratisl). 2025 Jan 16;80(3):561–72.

33. Ataide LMS, Moura CA, Fernandes JC, Araújo AB, Izumisawa CM, Figueiredo JVA, et al. Species composition and abundance of mosquitoes (Diptera: Culicidae) in a green area surrounded by urbanization in the Neotropical megacity São Paulo, Brazil. Entomological Communications. 2023 Dec 20;5:ec05041.

34. Borges MAZ, Silva HR da, Rodrigues CP, Santos CF, Souza GA de, Junior RR, et al. Biodiversidade de mosquitos (Diptera: Culicidae) em parques urbanos de Montes Claros – MG. Revista Unimontes Científica. 2024 Oct 29;26(2).

35. Heinisch MR, Medeiros-Sousa AR, Andrade PS, Urbinatti PR, Almeida RMMS, Lima-Camara TN. Fauna and Virological Investigation of Mosquitoes in Urban Parks in São Paulo, Brazil. J Am Mosq Control Assoc. 2023 Jun;39(2):75–84.

54. Hendy A, Hernandez-Acosta E, Valério D, Fé NF, Mendonça CR, Costa ER, et al. Where boundaries become bridges: Mosquito community composition, key vectors, and environmental associations at forest edges in the central Brazilian Amazon. PLoS Negl Trop Dis. 2023 Apr 26;17(4):e0011296.

55. Santos TP. Potential of Aedes albopictus as a bridge vector for enzootic pathogens at the urban-forest interfacein Brazil. Emerg Microbes Infect. 2018 Oct;

- Results are compared with relevant national and international literature. However, some recent publications from 2023–2024 could be more thoroughly discussed to reinforce the novelty and implications of the findings.

R.: Thank you for this valuable comment. As previously stated, we have revised the Discussion section to relevant studies published in 2023 and 2024.

- The study would benefit from a more explicit discussion on potential biases, particularly regarding the sampling duration and spatial coverage, which may influence species richness and virus detection rates.

R.: Thank you for your thoughtful comment. We have revised the Discussion section to explicitly address potential limitations of our study related to sampling duration and spatial coverage. The fllowing paragraph was added “Finally, it is important to acknowledge some limitations of our study. Mosquito collections were conducted during an active dengue outbreak and over a restricted temporal window (February to March 2024). These conditions may have influenced the observed species composition, infection rates, and viral circulation patterns, thereby limiting the generalizability of our findings to other time periods or non-outbreak contexts.”. Please, see lines 382-386.

- was there any report of DENV-1 in humans from that region?

R.: Yes, DENV-1 circulation in humans has been previously reported in the study region. We have now made this information clearer in the revised Discussion (lines 306–307) to contextualize our findings and support the relevance of vector-based viral detection.

- The conclusion could more explicitly highlight how the findings can inform future control strategies or surveillance efforts.

R.: Thank you for the suggestion. We have revised the Conclusion section to more clearly emphasize the practical implications of our findings. Please, see lines 396-401.

Minor Comments

- Abstract: Specify whether the study is observational, experimental, or integrative.

R.: Thank you for the observation. We have revised the abstract to explicitly state that this is an integrative study combining mosquito collection, arbovirus detection and sequencing, and ecological analyses. This clarification appears in the revised abstract (lines 29–30).

- Figures: Please ensure all figures are of sufficient resolution for publication. Some appear pixelated in the current version.

R.: Thank you for this comment. We have reviewed all figures; however, the submission process compresses their quality. We will provide high-resolution versions of all figures separately.

Several minor typos are present (e.g., ‘greater’ should be ‘higher’ in scientific context when referring to abundance).

* Line 296: ‘is higher to those’ should be corrected to ‘is higher than those’.

* Line 315: ‘eclectic in their blood-feeding habits’ – consider rephrasing to ‘eclectic blood-feeding behavior’.

R.: Thank you for your careful reading and helpful suggestions. “Greater” was replaced by “higher” through the manuscript;

Line 296 (now, 318) was corrected to “is higher than those”;

Line 315 (now, 336-337) was rephrased to “eclectic blood-feeding behavior” to improve clarity and style.

- Consider standardizing genus/species names in italics throughout the manuscript.

R.: Thank you for the suggestion. We carefully revised the entire manuscript to ensure that all genus and species names are consistently formatted in italics, in accordance with scientific conventions.

- Some acronyms such as UFF should be defined at first mention in both abstract and main text.

R.: Thank you for the observation. We have reviewed the manuscript and ensured that all acronyms, including UFF (Urban Forest Fragment), are now defined at their first mention in both the abstract and the main text.

- References: Consider including recent literature (post-2022) to reflect current studies in arbovirus-vector research.

R.: Thank you for this valuable suggestion. We have updated the reference list to include eight recent publications (from 2023 to 2025) that are relevant to arbovirus-vector research.

## Reviewer #2

Summary and General Assessment:

This manuscript presents an important and timely investigation into mosquito fauna and dengue virus circulation in urban and peri-urban environments during a dengue outbreak in Salinas, Minas Gerais, Brazil. The study is well-designed, combining entomological, virological, ecological, and genomic data. The integration of field collections, molecular detection of arboviruses, and phylogenomic analysis of DENV-1 provides a comprehensive approach. The conclusions are, in general, well supported by the results. The manuscript contributes significantly to the field of entomo-virology and public health surveillance, and I recommend it for publication after minor revisions.

R.: We sincerely thank the reviewer for the positive and encouraging assessment of our manuscript. We are glad that the study’s multidisciplinary approach and its contributions to entomo-virology and public health surveillance were appreciated. We have addressed the minor revisions suggested in the specific comments below to further improve the clarity and quality of the manuscript. Thank you again for your thoughtful review and recommendation for publication.

Minor Comments and Suggestions for Improvement:

1. Line 28 – “Many” sounds too strong

Suggest changing “Many species exhibit high levels of anthropophily…” to “Several species exhibit high levels of anthropophily…” as we do not have evidence for a large number of species with this trait.

R.: Thank you for the suggestion. We agree that “Several” is more appropriate in this context. The sentence has been revised to: “Several species exhibit high levels of anthropophily…” Please, see line 28.

2. Line 137 – Dissection procedure

A brief description of the head separation process should be included, especially to clarify measures taken to avoid cross-contamination during dissection.

R.: Thank you for this important observation. We have added a brief description of the dissection procedure to clarify the methodology and precautions taken to avoid cross-contamination. The revised sentence now reads: “Each mosquito’s head was carefully separated from its body under a stereomicroscope, using a sterile scalpel blade dedicated to that specimen to prevent cross-contamination.”. Please, see lines 147-149.

3. Line 138 – RNA extraction method

Although the authors refer to a previous study for the methodology, a brief summary of the RNA extraction procedure would help the reader and enhance reproducibility.

R.: Thank you for the suggestion. We have now added the following sentences to the manuscript: “Briefly, individual heads and pools of up to 10 bodies from the same species and sex were placed in enriched L-15 medium (20% fetal bovine serum, 0.5% non-essential amino acids, 1% penicillin, 0.1% gentamicin, and 0.1% fungizone). Tissues were homogenized with beads in a beadbeater for 30 s at 7500 rpm, followed by centrifugation at 12,000 rpm for 8 min at 4 °C. A 140 µL aliquot of the resulting supernatant was used for RNA extraction with the QIAamp Viral RNA Mini Kit (Qiagen), according to the manufacturer’s instructions.”. Please, see lines 150-156.

4. Line 214 – Clarify the procedure for positive pools

Please clarify that for pools that tested positive for dengue, RNA was extracted individually from mosquito heads and tested separately to confirm viral dissemination.

R.: Thank you for the observation. We have clarified this point in the revised manuscript. The sentence now reads: “For DENV-positive pools, RNA from each mosquito head was individually tested to confirm viral dissemination”. Please, see lines 159-160.

5. Line 237 – Figure font size

The font size in Figure 2 (phylogenetic tree) is quite small. Increasing the font size would improve readability, especially for the clade labels.

R.: Thank you for the suggestion. We have increased the font size of the labels in Figure 2 to the maximum possible, given space limitations. Please, see Figure 2 in the new manuscript version.

6. Line 273 – Verb tense in the Discussion

Although not incorrect, the use of the simple past tense in the Discussion is not ideal. The authors are encouraged to consider using the present tense when interpreting their findings, as is more standard in scientific discussions.

R.: Thank you for the observation. We have revised the Discussion section to use the present tense where appropriate.

7. Line 275 – Wording suggestion for caution

The phrase “with no other species implicated in the transmission” could be softened to “with no evidence of other species implicated in the transmission,” to reflect limitations of the detection methods.

R.: Thank you for the suggestion. We have adjusted the sentence to: “with no evidence of other species implicated in the transmission,” in order to more accurately reflect the limitations of our detection methods and avoid overstatement. Please, see line: 291.

8. Line 297 – Minimum infection rate (MIR) comparison

The values from the literature cited for MIR (e.g., 0.35 for Ae. albopictus and 3.37 for Ae. aegypti) should be more clearly contextualized, including the geographic and temporal setting of those previous studies for a more appropriate comparison.

R.: Thank you for this valuable observation. We have revised the text to provide clearer context for the MIR values cited. Specifically, we now state: “The minimum infection rate (MIR) for Ae. aegypti observed in this study is higher than those previously reported in the literature (MIR = 1.47 for adult Ae. albopictus collected in spring 2014 in São Paulo, Brazil, and 3.37 and 16.2 for immature and adult Ae. aegypti, respectively, collected in Rio Grande do Norte, Brazil, between 2011 and 2014).”. Please, see lines 317-321.

9. Line 298 – Study limitations and context

The discussion would benefit from a clearer framing of the characteristics and limitations of the study. Specifically, emphasize that the mosquito collections were conducted during an outbreak period and over a limited temporal window, which may influence the representativeness of the findings.

R.: Thank you for the suggestion. We have revised the discussion to include a clearer statement of the study’s limitations. The following paragraph was added: “Finally, it is important to acknowledge some limitations of our study. Mosquito collections were conducted during an active dengue outbreak and over a restricted temporal window (February to March 2024). These conditions may have influenced the observed species composition, infection rates, and viral circulation patterns, thereby limiting the generalizability of our findings to other time periods or non-outbreak contexts.”. Please, see lines 382-386.

Recommendation:

Minor Revisions

This manuscript is scientifically sound and relevant. I recommend it for publication pending minor revisions addressing the points above.

R.: We sincerely thank the reviewer for the positive evaluation and thoughtful comments that helped us improve the clarity, rigor, and impact of our manuscript. All suggested revisions were carefully addressed point by point, and the text was revised accordingly. We believe the updated version now meets the expectations for publication and we remain available for any further clarifications.

---

## [Reviewer Report · REVIEWERS COMMENTS]

## Reviewer #1

The authors have carefully addressed all the comments and suggestions raised in my previous review. The revised version of the manuscript has been significantly improved, and in my opinion, it now meets the requirements for publication. I recommend acceptance of the article in its current form.

## Reviewer #2

The authors have addressed and the concerns from my last review and the paper, now, is ready for publication.